# Common miRNAs of Osteoporosis and Fibromyalgia: A Review

**DOI:** 10.3390/ijms241713513

**Published:** 2023-08-31

**Authors:** Soline Philippe, Marine Delay, Nicolas Macian, Véronique Morel, Marie-Eva Pickering

**Affiliations:** 1Platform of Clinical Investigation Department, Inserm CIC 1405, University Hospital Clermont-Ferrand, F-63000 Clermont-Ferrand, France; philippesolinepro@gmail.com (S.P.); mdelay@chu-clermontferrand.fr (M.D.); nmacian@chu-clermontferrand.fr (N.M.); v_morel@chu-clermontferrand.fr (V.M.); 2Inserm 1107, Neuro-Dol, University Clermont Auvergne, F-63000 Clermont-Ferrand, France; 3Rheumatology Department, University Hospital Clermont-Ferrand, F-63000 Clermont-Ferrand, France

**Keywords:** fibromyalgia, osteoporosis, epigenetics, microRNA

## Abstract

A significant clinical association between osteoporosis (OP) and fibromyalgia (FM) has been shown in the literature. Given the need for specific biomarkers to improve OP and FM management, common miRNAs might provide promising tracks for future prevention and treatment. The aim of this review is to identify miRNAs described in OP and FM, and dysregulated in the same direction in both pathologies. The PubMed database was searched until June 2023, with a clear mention of OP, FM, and miRNA expression. Clinical trials, case–control, and cross-sectional studies were included. Gray literature was not searched. Out of the 184 miRNAs found in our research, 23 are shared by OP and FM: 7 common miRNAs are dysregulated in the same direction for both pathologies (3 up-, 4 downregulated). The majority of these common miRNAs are involved in the Wnt pathway and the cholinergic system and a possible link has been highlighted. Further studies are needed to explore this relationship. Moreover, the harmonization of technical methods is necessary to confirm miRNAs shared between OP and FM.

## 1. Introduction

Osteoporosis (OP) is a systemic disease that affects skeletal architecture. It is characterized by decreased bone mineral density (BMD) and increased risk of fragility fractures, disability, and impaired quality of life. OP prevalence increases with age at a rate of 19.1–23.5% in women after 50 years, and 5.9–7.2% in men, in whom it still remains underdiagnosed [1,2], and affects millions of people worldwide. OP is a complex multifactorial pathology that may remain undetected for a long time until a fracture occurs. It is diagnosed clinically and via dual-energy X-ray absorptiometry (DEXA). Mechanical, metabolic, and hormonal influences, aging, menopause, genetic predisposition, and environment, including nutrition, are the main factors for OP development. Although cost-effective therapeutic interventions to reduce fractures have been developed, there are a number of gaps in the general management of OP, and preventive measures need to be extended [3,4].

The fibromyalgia syndrome (FM) consists of chronic symptoms of moderate to severe intensity with chronic widespread (nociplastic) pain, associated with fatigue, cognitive and sleep disorders, and numerous somatic complaints [5,6,7,8]. Its manifestations are heterogeneous at the clinical, physical, social, and psychological levels, and treatment failures are frequent [9,10]. The prevalence of FM in the general population is estimated as 1.78% (95% confidence interval: 1.65–1.92) [11] with a female predominance, and, like in OP, it remains underestimated in men. The detection of FM follows the American College of Rheumatology recommendations [12], and, for its management, international guidelines recommend non-pharmacological approaches (exercise) in the first instance, and then drug treatment for comorbidities and pain, but quality of life is often impaired. Although a number of predictive factors of FM including biomarkers have been suggested [5], no specific biomarker has so far been identified.

Clinically, associations between FM and OP have been shown in the literature. Since the earlier works published in the 90s [13,14], FM has been suggested in a number of studies to be associated with an increased risk of OP [1,15,16]. In a meta-analysis, BMD at the lumbar spine was decreased in FM compared with normal individuals, stressing that the risk assessment of OP should be systematically performed [15]. FM shares common risk factors with OP, including age, gender, hygiene, dietary habits/lifestyle factors, a low level of physical activity, and hormonal factors [16]. A meta-analysis [17] demonstrated that BMD was significantly lower in FM patients, in Caucasians, and in female populations. A population-based case–control electronic study recently showed a significant association (coefficient correlation 0.55; *p* < 0.001) between OP and FM with a large database [1]. The authors underlined the need for the detection of predisposing factors for OP in FM patients, and advised the implementation of prevention measures (dietary supplements, resistance or weight-bearing exercise, anti-OP drugs). These, in order to maintain a satisfactory quality of life, reduce both the occurrence rate and severity of OP and its complications, such as fractures.

OP and FM have each been described as being associated with a number of possible biomarkers, including epigenetic markers and microRNAs (miRNAs) [18,19], known to play important roles in regulating gene expression. miRNAs have been a focus of research over the last years in OP and in FM. miRNAs, a class of non-coding RNAs 18 to 25 nucleotides long, are known to control gene expression at the post-transcriptional level [20] by forming an RNA-induced silencing complex which directly modulates the gene expression of mRNA genes [21]. miRNAs could regulate more than 60% of protein-coding genes and are therefore involved in most biological processes [22]. Epigenetics play important roles in bone metabolism and bone remodeling. An abnormal regulation may induce OP development and a number of miRNAs have been identified [23,24,25,26]. Likewise, miRNAs have been studied and described in FM [18,27]. Considering the need for specific biomarkers to improve OP and FM management, common miRNAs might provide promising tracks for future prevention and treatment. In order to find significant circulating predictive markers and potential new therapeutic targets for FM and OP, the objective of this review is to identify common miRNAs with similar regulation in OP and FM. To the best of our knowledge, there is currently no previous review on the common epigenetic markers of OP and FM.

## 2. Results and Discussion

### 2.1. Included Studies

Our search (Figure 1) retrieved 4922 potentially relevant records for the current scoping review, 232 for FM and 4690 for OP. After removing duplicates, the titles and abstracts of the remaining 1144 (36 for FM and 1108 for OP) were screened by two reviewers (SP, MD). After discarding records not conforming with the inclusion criteria, the full texts of the eligible papers were reviewed and 137 studies were included in this scoping review, 8 for FM [28,29,30,31,32,33,34,35] and 129 for OP [23,36,37,38,39,40,41,42,43,44,45,46,47,48,49,50,51,52,53,54,55,56,57,58,59,60,61,62,63,64,65,66,67,68,69,70,71,72,73,74,75,76,77,78,79,80,81,82,83,84,85,86,87,88,89,90,91,92,93,94,95,96,97,98,99,100,101,102,103,104,105,106,107,108,109,110,111,112,113,114,115,116,117,118,119,120,121,122,123,124,125,126,127,128,129,130,131,132,133,134,135,136,137,138,139,140,141,142,143,144,145,146,147,148,149,150,151,152,153,154,155,156,157,158,159,160,161,162,163].

In these 137 publications, there was a total of 189 dysregulated miRNAs (versus healthy volunteers (HV)): 46 miRNAs in FM and 166 in OP (Figure 2 and Figure 3, Appendix A).

In 45/137 articles, 23 miRNAs were common for FM and OP (Figure 2, Table 1): 14 articles showing 7 common miRNAs dysregulated in the same direction, and 40 articles showing 16 miRNAs dysregulated in the opposite direction (n = 9) or discordant (n = 7 dysregulated in the same or in the opposite direction in FM and OP).

In 14/45 articles, according to our objective to identify common miRNAs with a similar regulation, seven miRNAs were dysregulated (up- or downregulation) in the same direction for both pathologies: three miRNAs were upregulated (hsa-miR-9-(3p or 5p), hsa-miR-128-(3p), and hsa-335- 5p), and four were downregulated (hsa-miR-1-(3p), hsa-let-7a-(3p or 5p), hsa-miR-29a-3p, and hsa-miR-328-3p).

### 2.2. Characteristics of Studies with Common miRNAs in FM and OP, and With Regulation in the Same Direction

The 45 studies were published between 2012 and 2022, mainly with Caucasian participants for FM and Asian participants for OP (Appendix A).

#### 2.2.1. Sample Size

In the 14 articles related to the seven common miRNAs regulated in the same direction (up or down), three of the five FM studies had less than 50 participants, the largest (n = 74) with 49 FM patients and 25 HV [30], and the smallest (n = 18) with 10 FM and 8 HV [31]. For the nine articles for OP, two studies included more than 100 persons, four between 50 and 80 participants, and three less than 50. The largest study (n = 116) included 76 low-BMD participants with or without fractures and treatment, and 40 HV [43], and the smallest (n = 6) included 3 OP and 3 HV [37].

In the 45 articles, the majority of FM studies (71%: n = 5) had less than 50 participants, the largest (n = 74) with 49 FM patients and 25 HV [30], and the smallest (n = 18) with 10 FM and 8 HV [31]. For OP, six studies included more than 100 persons, six between 50 and 80 participants, and twenty-five less than 50. One [55] did not mention the sample size. The largest study (n = 161) included 82 OP and 79 HV [65], and the smallest (n = 6) 3 OP and 3 HV [37,47].

#### 2.2.2. Age and Gender

Concerning the seven miRNAs regulated in the same direction (up or down), all cohorts in FM patients were around 50 years old, except one [28] that did not mention the age. In OP, the patients were over 50 years old, with 70-year-old persons in five studies [23,36,37,41,43] and no mention of age in several [38,39,40]. In FM, all studies but one [32]—with only 30% men—included only females. In OP, six out of nine studies for OP were 100% women. One study included both genders, with 60% women [42]. The remaining two studies did not mention gender [39,41].

In the 45 articles, the FM patients were around 50 years old, and OP patients were over 50 years old, with 80-year-old persons in 2 studies [56,59]. Age was not always mentioned in FM [28] or in OP [38,39,40,46,48,55,70,72]. In FM, all studies but one [32]—with only 30% men—included only females. In OP, 25 of 38 studies for OP had only women. For studies that included both genders, the majority was with 60 to 90% women. One study included the same number of men and women for patient and HV groups [56]. The remaining nine studies did not mention gender.

#### 2.2.3. Tissue Sample, Extraction, and Detection Method

For the seven miRNAs regulated in the same direction (up or down), concerning extraction, four out of five FM studies were on circulating miRNAs: serum [28,29,32], cerebrospinal fluid [31], or whole blood [30]. Three of the studies used extraction kits from Qiagen, one from Thermo Fisher, and one did not specify the kit used. Concerning detection, qRT-PCR was used, but, in one study [32], a multiplex assay was used. Concerning extraction in OP, 29% of OP studies (n = 4) used circulating miRNAs with serum (n = 4). Other studies used whole blood (n = 3), or other tissues specific to bone, like human-bone-marrow-derived mesenchymal stem cells (hBMSCs) (n = 1) and bone (n = 1). In total, 67% (n = 6) of the studies used extraction kits from Thermo Fisher, the other articles used Qiagen. Three of the four articles with sequencing are found in the nine OP articles where the regulation is in the same direction. A total of 67% (n = 6) used qRT-PCR as a detection method.

In the 45 articles, the extraction was realized with Qiagen or Thermo Fisher (Invitrogen, Ambion) kits, and the detection method was qRT-PCR. Concerning extraction in FM, 57% of FM studies (n = 4) were on circulating miRNAs in the serum [28,29,32], cerebrospinal fluid [31], white blood cells [33], peripheral blood monocellular cells [34], and whole blood [30]. Overall, 57% (n = 4) of the studies used extraction kits from Qiagen, two studies used Thermo Fisher, and one did not specify the kit used. Concerning detection, qRT-PCR was used, but one study [32] used a multiplex assay.

Concerning extraction, in OP, 55% of OP studies (n = 21) used circulating miRNAs, with serum (n = 15) and plasma (n = 6), or whole blood (n = 5) and circulating monocytes (n = 1). Other tissues specific to bone were used like human-bone-marrow-derived mesenchymal stem cells (hBMSCs) (n = 6) and bone (n = 3); two studies [56,59] used both bone and serum. In total, 53% (n = 20) of the studies used extraction kits from Thermo Fisher, 42% (n = 16) used Qiagen, one study used both [37], and one did not specify the kit used [61]. Overall, 86% (n = 33) of OP studies used qRT-PCR as the detection method, only four studies performed sequencing, and one study performed next-generation sequencing (NGS) [53].

### 2.3. Discussion

Low BMD and OP have been shown to be linked to FM [1,13,14,15,16,17]. In order to better characterize the association between both pathologies, the aim of this review was to identify if there are miRNAs common to FM and OP. It also aimed to specify which miRNAs are regulated in the same direction, and to suggest common biomarkers.

A total of 23 common miRNAs were retrieved in the literature. Fifteen of these are described to target and modulate the Wingless integration site (Wnt) pathway (miR-1-(3p), let-7a-(3p or 5p), miR-9-(3p or 5p), miR-21-5p, miR-29a-3p, miR-107, miR-133a, miR-139-5p, miR-145-(3p or 5p), miR-148a-(3p), miR-186-5p, miR-320a, miR-328-3p, miR-335-5p, and miR-338-3p) [164,165,166,167]. Seven miRNAs modulate the cholinergic system (miR-7-5p, miR-9-(3p or 5p), miR-128-(3p), miR-148a-(3p), miR-186-5p, miR-328-3p, and miR-532-(3p or 5p)), known as “CholinomiR” [30,168]. It is interesting to note that three miRNAs are upregulated and four downregulated in FM and in OP, and they target the Wnt pathway in bone studies, and the cholinergic system in FM studies.

The Wnt system is composed of Wnt proteins that are involved in many cellular processes, ranging from cell-fate determination to stem-cell renewal, and dysregulated Wnt signaling is involved in many human pathologies [169]. Wnt pathways display numerous cross connections that negatively or positively regulate each other, forming a mutual regulatory network.

At the bone level, the Wnt pathway is the most important regulatory pathway; it has a direct effect on skeletal remodeling, regulating bone mass via bone forming osteoblasts, old bone reabsorbing osteoclasts, and progenitor cells responsible for the maintenance of bone-marrow-derived mesenchymal stem cells (BMSCs) [170]. Multiple genes are also targeted, and involved in the regulation of bone, including cyclin D1, RUNX2, bone sialoprotein, sclerostin, Dickkopf 1 and 2, secreted FZ-related protein, osteoprotegerin, osterix, myocyte enhancer factor 2C, osteocalcin, or osteopontin [165,170]. Wnt signaling plays a central regulatory role during embryonic development and in the adult osteogenic differentiation of mesenchymal stem cells. Alterations of this system are accompanied by impaired bone healing, autoimmune disease, osteoporosis, and malignant degeneration. Wnt factors have been suggested as potential future therapeutics to help bone healing after trauma in endocrine or orthopedic situations [171].

The Wnt pathway is ubiquitous and involved in other domains like muscle function [172] or chronic fatigue syndrome (CFS) [173]. FM and CSF are flip sides of almost identical chronic conditions. Patients with CFS have many similar symptoms to those with fibromyalgia—brain fog, fatigue, headache, and poor sleep. Dysregulated Wnt/β-catenin signaling has been shown to cause oxidative stress in animals with CFS, and produce reactive oxygen species (ROS) and aberrations in cross-talks between Wnt, Redox, and NF-kB pathways. The Wnt pathway is also involved in pain sensitization, and neuropathic and bone-cancer-induced pain [174]. Pain accompanies OP especially after trauma and fracture; ensuing central sensitization may lead, in some patients, to neuropathic pain (with burning, stabbing, itching, allodynia, and hyperalgesia). While the origin of FM remains unclear, the central sensitization of pain [175] and the diffuse nociplastic musculoskeletal type of pain are landmarks of FM [6]. In a study centered on an experimental model of FM [174], it has been shown that the Wnt/β-catenin pathway is involved in the release of brain-derived neurotrophic factor from the spinal microglia. This observation suggests that the modulation of this pathway plays a key role in the activation of the nociceptive pathway in the spinal cord [174]. In the same way, the activated Wnt signaling pathway in neuropathic pain [176] modulates the expression of the glutamate receptor, resulting in synaptic plasticity and central sensitization [176].

The cholinergic system is the other system that is expressed by the common miRNAs in FM and OP. Cholinergic mechanisms may play an important role in the pathophysiology and severity of FM [30], especially via the vagus nerve. The vagus nerve is a major cholinergic component of the parasympathetic system, a mixed nerve containing 80% afferent and 20% efferent fibers, which controls the neuro-digestive, vascular, and immune systems. Non-invasive vagus nerve stimulation is even considered today as a potential adjunct treatment for FM [177], since FM involves a dysregulation of the autonomic (high sympathetic tone) and immune (enhanced pro-inflammatory activity and cytokines) systems. Another link between FM and the cholinergic system is medication: a cross-sectional study [178] highlighted that one of the most frequently used and effective FM treatments is amitriptyline, a strong anticholinergic molecule [178].

The cholinergic system is also involved in the health status of bone [179], and cholinergic components (with adrenergic ones, the other branch of the autonomic nervous system) play an important role in bone remodeling. Bone loss associated with OP could be due to local alterations/inhibitions in cholinergic activity, but this has been scarcely studied so far. Clinical studies have shown that bone is also associated with the function of cholinergic-regulated tissues like the hypothalamus [180] and those outside the nervous system in non-neuronal cells.

Common miRNAs have specific signatures in FM and OP (Figure 4, Table 2).

miR-1-(3p), involved in myoblast differentiation, has been suggested to be downregulated in FM because of decreased physical activity in this pathology. miR-1(3p) also modulates Brain Derived Neurotrophic Factor (BDNF) expression in skeletal muscle where it inhibits myogenic differentiation [28,37], and in OP [36]. miR-1-(3p) has a specific target with Secreted Frizzled Related Protein 1 (SFRP1), part of the Wnt signaling system, and balances the osteogenesis and adipogenesis of mesenchymal stem cells (MSCs). Overall, downregulated miR-1-(3p) leads to decreased bone formation and diminished myogenesis [28,36,37].

let-7a-(3p or 5p), downregulated in both diseases [29,38,41,73], plays a role in nerve fiber pathology [35] and regulates pain pathways via the endogenous opioid system in FM [29]. It has a close association with the Wnt pathway in OP [37,41].

miR-9 (3p or 5p) is upregulated in both pathologies. As a cholinomiRNA, it may shift inflammatory processes (Janus kinase 2 expression)—possibly linked to pain—via the modulation of the systemic cholinergic system in FM [30]. It binds to the Wnt pathway, and has a deleterious effect on bone quality [38], skeletal cell proliferation, and differentiation [38,39].

miR-29a-3p is downregulated and is associated with the diminution of β-catenin and with the inactivation of canonical Wnt signaling, leading to OP [23]. In FM, miR-29a-3p expression is reduced compared to healthy controls [31].

miR-128-3p, which is upregulated, is an inhibitor of bone formation via sirtuine 6 (SIRT6) expression [40], and is a cholinomiRNA [30] that modulates the systemic cholinergic system in FM. miR-328-3p is also a cholinomiRNA, which is downregulated, and is associated with osteoblast differentiation in OP [41].

mir-335-5p, which is upregulated, is the only miRNA that was significantly expressed in 105 FM patients compared to 54 controls [32]. mir-335-5p activates Wnt signaling and promotes osteogenic differentiation by downregulating Dickkopf 1 (DKK1), a soluble inhibitor of the Wnt signaling pathway. Two studies with 26 and 39 OP patients compared to control subjects showed miR-335-5p as being upregulated in OP with vertebral/fragility fractures [42,43].

With these seven common miRNAs up- or downregulating in the same direction in FM and OP, the possibility of a link between the Wnt pathway and the cholinergic system could open new avenues for research on the prevention and management of both pathologies. Preclinical studies have highlighted a cholinergic induction of Wnt during infection or immune activation [181,182]. The coordinated activities between acetylcholine receptors and Wnt signaling seem to be conserved in evolutionary terms, and are found in mice [183]. There seems to be a cholinergic–Wnt signaling axis, which can intervene in homeostasis regulation [181], but has not been looked for in OP or FM.

Although we highlighted a number of miRNAs common in FM and OP, there are a number of limitations in this review. The regulation of miRNAs may often be contradictory, even in the same pathology. These mismatches can be explained by differences between studies. From one study to another, the number of patients in the cohort but also in the tissue samples, and technical issues with extraction and detection techniques all vary. miRNAs in FM appear to have less contradictions than in OP, but less research has been carried out on this topic. We noticed that the more studies report miRNA expression, the more contradictory results are obtained.

Another limitation in the interpretation of our results is the ethnical origin of the patients. Indeed, the majority of articles included Asian patients, and results cannot be extrapolated as representing Caucasian characteristics; in addition, there are no studies in African patients or other ethnical groups. While OP is largely reported in women, especially after menopause, OP remains underestimated in men where it usually presents at a later age. Likewise, mostly women seek advice for FM symptoms, while men are less present in pain clinics for this pathology and are poorly represented in publications on FM. This present review underlines the paucity of miRNA studies that include men; such a gap needs to be addressed, since *gender*-specific differences in *miRNAs* expression (in the quantity of expression and type of miRNAs) have been described in several pathologies, and such a difference in OP and FM could suggest the development of sex-specific therapeutic strategies.

There is also a need to have prospective longitudinal rather than transversal studies in order to follow FM patients over a number of years, and to evaluate, thoroughly, their bone health and the early presence of OP. Future clinical research should include larger cohorts of patients with a wider range of ethnic representation, more men, and a systematic report on comorbidities that could influence the given pathological condition. Bioinformatics and artificial intelligence can be useful tools in this context to validate miRNAs and identify the predictive value of these common biomarkers for the better diagnosis and management of both pathologies. These miRNAs play a pivotal role in the pathogenesis of FM and OP and are associated with cardinal symptoms, making them interesting potential therapeutic molecules to target.

Finally, some miRNAs are not detected in both pathologies, but it is not certain whether they are present or not, or shared between both diseases, as miRNAs are often prescreened and the choice is based on what researchers want to analyze [184]. Some miRNAs are not secreted by cells, and therefore cannot be detected in circulating miRNAs studies. While qRT-PCR is a sensitive and specific reference method that allows the detection of individual miRNAs or a panel of miRNAs, new detection methods now exist, including deep sequencing. These may help in the future to detect a much wider range and identify new miRNAs, but such techniques are expensive and their development is self-limiting.

## 3. Materials and Methods

### 3.1. Review Protocol

This is a systematic/scoping review that was conducted and reported according to the Preferred Reporting Items for Systematic Reviews and Meta-Analyses extension for Scoping Reviews (PRISMA-ScR) Guidelines [185].

### 3.2. Eligibility Criteria

Peer-reviewed journal articles were included if they involved the comparison of miRNA expression between HV and patients with OP or FM. An investigation of the miRNAs profiles was followed as a primary or secondary outcome, with any type of biological sample. Clinical trials, case–control, and cross-sectional studies were included. Reviews, meta-analyses, comments, method validations, meeting abstracts, in vitro studies, and animal studies were excluded. All articles that did not clearly mention FM, OP, or miRNA expression were excluded. Database-based studies were not included. There was no age limit nor a minimum number for the population, no specific requirement regarding the year of publication, and studies had to be available in English.

### 3.3. Information Sources and Search Strategy

In order to identify potentially relevant articles, the online database PubMed was searched until June 2023 with the following keywords: (“fibromyalgia” OR “fibromyalgia syndrome” OR “osteoporosis” OR “post-menopausal osteoporosis”) AND (“miRNA” OR “microRNA” OR “sequencing microRNA” OR “micro RNA” OR “mirRNAs OR “mirs” OR “mir”). The gray literature was not searched nor included. The medical subject heading (MeSH) was used to increase the sensitivity of the systematic search. The reference list of all the full-text articles selected after the screening and the list of articles citing these articles were hand-searched for titles not identified with the previous methods.

### 3.4. Study Selection Process

Abstracts were obtained for all the studies identified during the electronic and hand-searches, after having removed duplicates. Two reviewers (SP, MD) screened titles and abstracts in the first phase, and full-text copies in the second phase independently to eliminate articles that clearly failed to meet the eligibility criteria. Full-text copies were obtained for all the selected studies.

### 3.5. Data Charting Process and Synthesis

Predetermined data (including first author, publication year, number for the population, ethnicity, mean age, gender, menopause, age of the disease, comorbidities, biological sample, extraction kit, detection method, study design...) were extracted from each study by two reviewers (SP and MD) and arranged into data tables. The miRNAs cited and explored in the included studies were listed in tables specifying their expression (upregulating or downregulating), according to the targeted pathology (FM or OP). No quality assessment of the included articles took place, which was in accordance with the available guidelines on scoping reviews [185]. We used RNAcentral (http://rnacentral.org), miRBase (http://www.mirbase.org), and Rfam (http://rfam.org) databases (accessed on 18 May 2023) to identify signaling pathways and systems potentially altered by the miRNAs differentially expressed in both pathologies (FM and OP).

According to the nomenclature of the miRNAs, we paid attention to miRNAs originating from the same precursor, the most abundant being named “5p” and the least abundant “3p”. Some articles did not, however, specify the strand. If the literature reported one or the other strand, we indicated it (3p or 5p).

## 4. Conclusions

Collective data of this review show that a number of common miRNAs in FM and OP have been identified. These are involved in the Wnt pathway for OP and in the cholinergic system for FM. A substantial link is still missing to evaluate the real miRNA impact on Wnt dysregulation in FM and on cholinergic system alterations in OP. Research on this possible link is important since it has been described in animal species with intestinal dysfunction and could be investigated in musculoskeletal conditions. Further research is also warranted on harmonizing techniques or on the choice of tissue for miRNA analysis. Our review underlined a number of gaps linked to the large heterogeneity of methods. We recommend further studies in order to strengthen the epigenetic knowledge of the FM and OP association, and how they are interwoven, to prevent and better manage these pathologies using common predictive biomarkers. Clinical data have reported the increased risk of developing OP in FM patients. The identification of common miRNAs would provide predictive factors for limiting the double burden of FM and OP in aging.

## Figures and Tables

**Figure 1 ijms-24-13513-f001:**
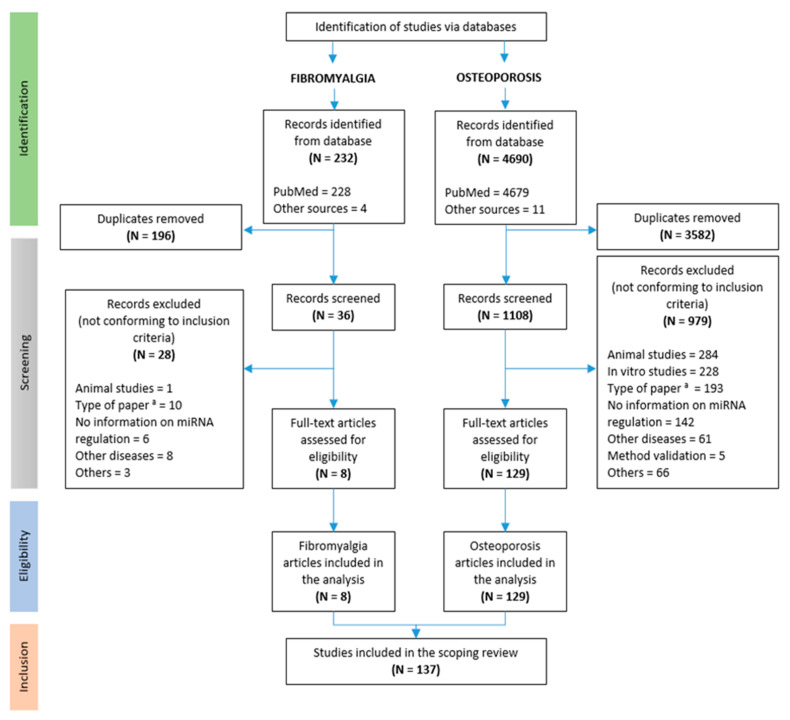
PRISMA Extension for Scoping Reviews (PRISMA-ScR) flowchart. ^a^: comment, meta-analysis, review, meeting abstract; N = number of articles.

**Figure 2 ijms-24-13513-f002:**
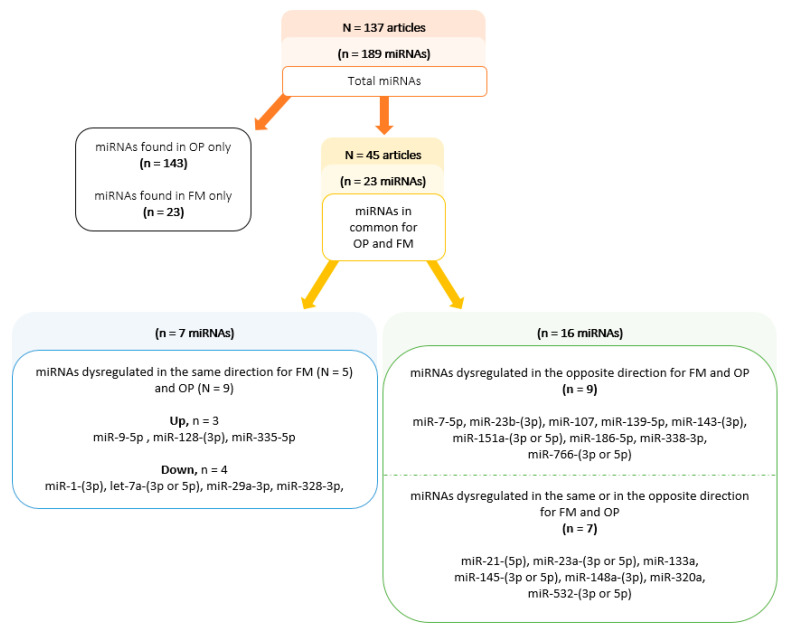
Summary diagram of dysregulated miRNAs in both osteoporosis (OP) and fibromyalgia (FM). N = number of articles; n = number of miRNAs.

**Figure 3 ijms-24-13513-f003:**
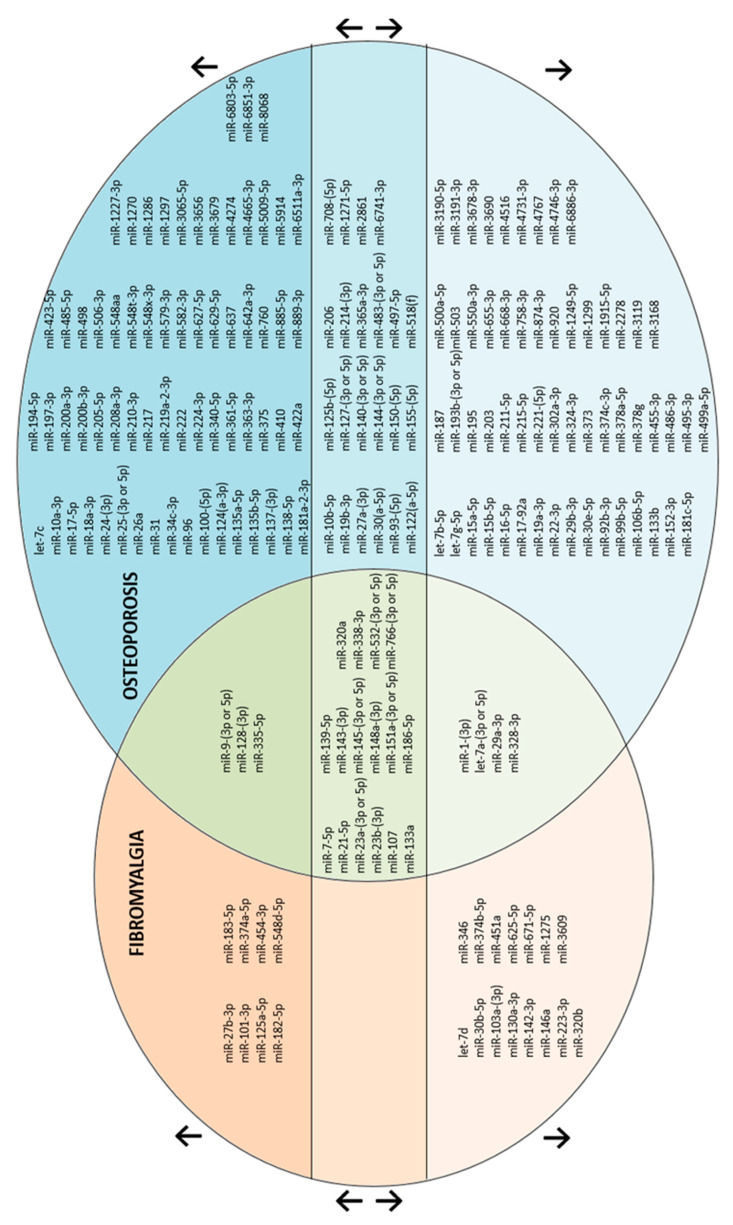
Summary of miRNAs found in fibromyalgia (orange circle) and in osteoporosis (blue circle). “↑” miRNAs at the upper part of the circles are upregulated. “↓” miRNAs at the lower part of the circles are downregulated. “↑↓” miRNAs at the equator of the circles are either up- or downregulated. Intersection of circles in green represents miRNAs common for FM and OP. References are listed in Appendix A.

**Figure 4 ijms-24-13513-f004:**
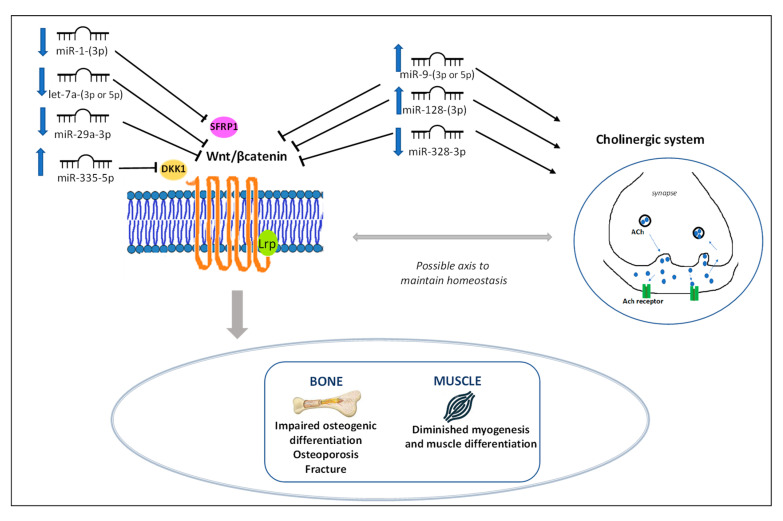
Common miRNAs in fibromyalgia and in osteoporosis and regulation via the Wnt signaling pathway and/or the cholinergic system. References are listed in Table 1 and Table 2. (↓ = downregulated; ↑ = upregulated).

**Table 1 ijms-24-13513-t001:** Differentially expressed miRNAs in fibromyalgia (FM) and osteoporosis (OP). (↓ = downregulated; ↑ = upregulated).

miRNAs Expressed in Same Direction in FM and OP (n = 7).				miRNAs Expressed in Opposite Direction in FM and OP (n = 9)			
miRNAs	miRNA Expression		References	miRNAs	miRNA Expression		References
**miR-1-(3p)**	**↓**	**FM**	[28]	**miR-7-5p**	**↑**	**FM**	[30]
		**OP**	[36,37]		**↓**	**OP**	[42]
**let-7a-(3p or 5p)**	**↓**	**FM**	[29]	**miR-23b-(3p)**	**↓**	**FM**	[31]
		**OP**	[37]		**↑**	**OP**	[44,45]
**miR-9-(3p or 5p)**	**↑**	**FM**	[30]	**miR-107**	**↓**	**FM**	[29,33]
		**OP**	[38,39]		**↑**	**OP**	[46]
**miR-29a-3p**	**↓**	**FM**	[31]	**miR-139-5p**	**↓**	**FM**	[28]
		**OP**	[22]		**↑**	**OP**	[44]
**miR-128-(3p)**	**↑**	**FM**	[30]	**miR-143-(3p)**	**↓**	**FM**	[34]
		**OP**	[40]		**↑**	**OP**	[41,47]
**miR-328-3p**	**↓**	**FM**	[30]	**miR-151a-(3p or 5p)**	**↓**	**FM**	[29]
		**OP**	[41]		**↑**	**OP**	[48,49]
**miR-335-5p**	**↑**	**FM**	[32]	**miR-186-5p**	**↑**	**FM**	[30]
		**OP**	[42,43]		**↓**	**OP**	[42]
				**miR-338-3p**	**↓**	**FM**	[34]
					**↑**	**OP**	[50]
				**miR-766-(3p or 5p)**	**↓**	**FM**	[30]
					**↑**	**OP**	[51]

**Table 2 ijms-24-13513-t002:** Characteristics of the 7 common miRNAs in fibromyalgia (FM) and osteoporosis (OP).

miRs	Pathology	Reference	Findings
**miR-1(-3p)**	**FM**	[28]	**Myoblast differentiation** - *Downregulation of miR-1(-3p) following reduced physical activity in FM patients because of pain and fatigue symptoms.*
			**Modulation of brain-derived neurotrophic factor (BDNF) expression.** *BDNF in skeletal muscle inhibits myogenic differentiation.*
	**OP**	[36]	**Balance between osteogenesis and adipogenesis of mesenchymal stem cells (MSC).** *Upregulation of miR-1-3p during osteogenesis and downregulation during adipogenesis. Secreted Frizzled-related protein 1 (SFRP1) is a direct target of miR-1-3p. Inhibition of miR-1-3p decreased bone formation and bone mass.*
			**Regulation of myostatin gene.** *Influence on muscle hypertrophy.*
			**Reduced/suppressed expression of fibronectin 1, BDNF, Dickkopf 1 (DKK1).** *Suppressed proliferation and migration of oral squamous cell carcinoma/ renal cell lines.*
		[37]	**Regulation of RAB5C expression.** *hsa-miR-1-3p and his target mRNA RAB5C may play a critical role in the bone metabolism of postmenopausal osteoporosis.*
			**Regulation in skeletal tissue.** *Muscle proliferation, muscle differentiation and myogenesis.*
**let-7a(-3p or 5p)**	**FM**	[29]	**Repression of μ-opioid receptor expression.** *Regulation of the endogenous opioid system and in opioid tolerance.*
	**OP**	[37]	**Downregulated.**
**miR-9(-3p or 5 p)**	**FM**	[30]	**CholinomiRNA. Regulation of Janus kinase 2 (JAK2) expression. JAK2 is pivotal for IL6/JAK2/STAT3 axis-mediated inflammation.** *CholinomiRs may shift inflammatory processes via modulation of the systemic cholinergic system.*
	**OP**	[38]	**Inhibition of osteogenic differentiation of hMSCs.** *Inhibiting the miR-9-5p expression promoted the expression of osteocalcin, runt-related transcription factor 2 (Runx2) and bone morphogenetic protein 7 (BMP-7), enhanced BMD, and promoted fracture healing.*
			**Inhibition of skeletal cell proliferation and differentiation.**
		[39]	**Inhibition of the expressions of osteogenic-related genes.** *Direct binding to Wnt3a. Wnt3a overexpression partially reversed the regulatory effect on osteogenic differentiation of MSCs.*
			**Promotion of adipogenic-related genes expression.**
**miR-29a-3p**	**FM**	[31]	**Upregulation during aging in mice and decreased in several pathologies including muscular dystrophy type 1.**
	**OP**	[23]	**Induction of beta-catenin protein levels.** *Activation of canonical Wnt signaling.*
			**Key regulation of collagen expression.**
**miR-128-3p**	**FM**	[30]	**CholinomiRNA- Upregulation and Involvement in the neuronal oxidative stress response.** *Modulation of the cholinergic system.*
	**OP**	[40]	**Inhibition of osteoblast differentiation.** *Down-regulation of sirtuine 6 (SIRT6) expression.*
**miR-328-3p**	**FM**	[30]	**CholinomiRNA.** *Modulation of the cholinergic system.*
	**OP**	[41]	**Inhibition of osteoblast differentiation.**
**miR-335-5p**	**FM**	[32]	**Sole miRNA that differed significantly from controls.**
	**OP**	[42]	**Upregulation in osteoporosis with low-traumatic fractures compared to controls.**
		[43]	**Upregulation in osteoporosis with vertebral fractures/low BMD compared to low BMD/no fractures and controls.**

## Data Availability

The original data of this present study are available from the corresponding authors.

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
