# Peer review of "Common miRNAs of Osteoporosis and Fibromyalgia: A Review"

_ijms, 2023, doi:10.3390/ijms241713513_

Round 1
Reviewer 1 Report
In this systematic review, the authors try to identify common miRNAs in OP and FM which is an interesting work. However, there are some major problems.
1. I could not find the PROSPERO information (registration number) on this manuscript. On the fundamental knowledge, before making the systematic review manuscript, the researchers need to register it with PROSPERO (International prospective register of systematic reviews) on an individual basis.
2. It was recommended provide a studies list which included in your review.
3. The authors found 7 common miRNAs and Wnt pathway may play a great role in OP and FM. Are these 7 miRANs possessed specific effects in Wnt pathway? More discussion should be provided.
4. It was also recommended provided some graphical explanation based on your finding, especially their association and effects with Wnt pathway.
The quality of English is ok.
Author Response
Dear Reviewer,
We thank the reviewer for his/her remarks and suggestions and reply point to point to his/her comments. We hope these will be satisfactory. Yours sincerely.
Comments and Suggestions for Authors
In this systematic review, the authors try to identify common miRNAs in OP and FM which is an interesting work. However, there are some major problems.
- I could not find the PROSPERO information (registration number) on this manuscript. On the fundamental knowledge, before making the systematic review manuscript, the researchers need to register it with PROSPERO (International prospective register of systematic reviews) on an individual basis.
The review was originally planned as a scoping review and it was not necessary in that case to register, as we recently experienced with a scoping review in European Journal of Pain Journal (Osteoporosis treatment and pain relief: A scoping review. Pickering ME et al. Eur J Pain. 2023 Jul 5. doi: 10.1002/ejp.2156. PMID: 37403555).
Unfortunately, for the present review, it was too late to get registration once we had almost finished. We are really sorry about this mishap as we are usually very rigourous to declare our work before starting work like in clinical trials. We will be careful for future publications.
- It was recommended provide a studies list which included in your review.
The list of studies that has been included in the analysis is presented as a supplementary file (File S5)
- The authors found 7 common miRNAs and Wnt pathway may play a great role in OP and FM. Are these 7 miRANs possessed specific effects in Wnt pathway? More discussion should be provided.
Thank you for your remark. We have enriched the discussion according to the reviewer’s suggestions page 9-10, added Table S6 and a figure (Figure 4).
- It was also recommended provided some graphical explanation based on your finding, especially their association and effects with Wnt pathway.
We have proposed a graphical presentation according to the reviewer’s suggestions (Figure 4).

Reviewer 2 Report
Studies have confirmed that Fibromyalgia is one of the important risk factors for osteoporosis, but the exact mechanism is unknown. The main purpose of this paper is to summarize the common miRNAs of Fibromyalgia and osteoporosis. Specific suggestions are as follows:
1. Title: "Osteoporosis and Fibromyalgia: A systematic review of common miRNAs" is proposed to be revised to “common miRNAs of Osteoporosis and Fibromyalgia: A systematic review”.
2. Simply listing miRNAs associated with Osteoporosis and Fibromyalgia is not of sufficient scientific importance. It is more important to deeply analyze the function of each miRNA, relationships with diseases and future research value
3. Bioinformatics analysis or simple in vitro experiments are recommended to validate the screened miRNAs if possible.
Moderate editing of English language required.
Author Response
Dear Reviewer,
We thank the reviewer for his/her remarks and suggestions and reply point to point to his/her comments. We hope these will be satisfactory. Yours sincerely.
Comments and Suggestions for Authors
Studies have confirmed that Fibromyalgia is one of the important risk factors for osteoporosis, but the exact mechanism is unknown. The main purpose of this paper is to summarize the common miRNAs of Fibromyalgia and osteoporosis. Specific suggestions are as follows:
- Title: "Osteoporosis and Fibromyalgia: A systematic review of common miRNAs" is proposed to be revised to “common miRNAs of Osteoporosis and Fibromyalgia: A systematic review”.
This has been changed according to the reviewer’s suggestions.
- Simply listing miRNAs associated with Osteoporosis and Fibromyalgia is not of sufficient scientific importance. It is more important to deeply analyze the function of each miRNA, relationships with diseases and future research value.
Thank you for these suggestions to improve our manuscript. We have added a supplementary analysis of the miRNAs for the involvement of each miRNA page 9-10, a supplemental Table S6, and included a graphical representation (Figure 4). We also discussed future research value of miRNAs page 11.
- Bioinformatics analysis or simple in vitro experiments are recommended to validate the screened miRNAs if possible.
Thank you for this suggestion. For this review we did not have the means (and the finance...) to benefit from such analyses.
We follow however at the moment 2 large cohorts, on fibromyalgia (FIDGIS NCT04624581) and osteoporosis (VASCO NCT05228262 and MAGELLAN NCT05328154) for which we obtained a grant that will allow us to genotype miRNAs and use in vitro analysis and bioinformatics analysis.
The paper has been edited by an English native speaker and corrections appear in yellow in the text.

Reviewer 3 Report
A significant clinical association between osteoporosis (OP) and fibromyalgia (FM) has been shown in the literature. Given the need for specific biomarkers to improve OP and FM management, common miRNAs might provide promising tracks for future prevention and treatment. The aim of this review is to identify miRNAs described in OP and FM, and dysregulated in the same direction in both pathologies. Collective data of this review show that a number of common miRNAs in FM and OP have been identified. These are involved in the Wnt pathway for OP and in the cholinergic system for FM, but a thread is missing to evaluate the real miRNAs impact on Wnt dysregulation in FM and of the cholinergic system alterations in OP.
Introduction. The authors make a description of the both diseases and establish the hypothesis
The methodology is complete, widely described, which would allow the study to be carried out by another research group.
The results are clear expressed in tables and easy to understand
The discussion is adapted to the results obtained. The authors express the limitations and strengths of the study.
Author Response
We thank the reviewer for his/her review and encouraging remarks.

Round 2
Reviewer 1 Report
According to the responses from authors, it is suggested that the title should delete 'systematic'.
The quality of English language is OK.
Author Response
According to the reviewer's suggestion the word systematic has been removed from the title.
Thank you for your reviewing
Dr ME Pickering
Reviewer 2 Report
Table S6 can be included into the main text.
None
Author Response
According to the reviewer's suggestion, Table S6 has been included in the main text as Table 2
Thank you for your reviewing
Dr ME Pickering